# Variation in spend on young mental health across Clinical Commissioning Groups in England: a cross-sectional observational study

Stephen Rocks,[1] Mina Fazel,[2] Apostolos Tsiachristas[1]

¹Health Economics Research Centre, University of Oxford, Oxford, UK
²Department of Psychiatry, University of Oxford, Warneford Hospital, Oxford, UK

**Correspondence to**
Stephen Rocks;
stephen.rocks@ndph.ox.ac.uk

## ABSTRACT

**Objectives** To investigate whether the rate of spend on child and adolescent mental health is influenced by demand for other competing services in local commissioning decisions.

**Design** Analysis of spend data by Clinical Commissioning Groups (CCG), including other publicly available data to control for variation in need.

**Setting** Local commissioning decisions in the National Health Service.

**Participants** Commissioning of health services across 209 CCGs.

**Main outcome measures** Association between the rate of child and adolescent mental health spend and demand for child and adolescent mental health services (CAMHS), adult mental health services and physical health services after adjusting for confounding factors.

**Results** An additional percentage point in the proportion of children in care is associated with 4% higher child and young person mental health (CYP MH) spend per person aged 0–18 (ratio of means: 1.04; 95% CI 1.00 to 1.07). Spending £100 more on physical health services was associated with 9% lower spend in CYP MH per person aged 0–18 (ratio of means: 0.91; 95% CI 0.84 to 0.99).

**Conclusions** Healthcare commissioners in England face a challenge in balancing competing needs. This paper contributes to our understanding of this by quantifying the possible extent of the trade-off between physical health and CYP MH when allocating budgets. Any attempt to explain the variation in CAMHS spend must also take account of demand for other services.

## INTRODUCTION

Mental health problems impose a significant burden on individuals, their families and society.[1] It is estimated that the costs associated with mental illness in the UK total 4.5% of gross domestic product, driven mainly by productivity loss and opportunity cost of informal care.[2] Childhood and adolescence is crucial to any attempt to alleviate this burden because the majority of lifelong mental illness has started by the age of 17,[3] exerting short-term and long-term costs that impact significantly across the life course.[3] However,

### Strengths and limitations of this study

► This is the first study looking broadly at the factors associated with the commissioning of child and adolescent mental health services.
► A major strength is the incorporation of different area-level data.
► There are limited data available for spend at a commissioning level.
► There are concerns over the consistency of published figures for spend on child and young person mental health.
► The analytical approach and incorporation of sensitivity analyses strengthen this study.

although effective interventions exist,[4] most people do not access mental health interventions during their childhood and adolescence.[5 6] It is estimated that less than a quarter of young people with a mental health problem receive any help from specialist services.[5] This treatment gap reflects a disparity between the needs and resources committed to the mental health of young people.

In England, the decision on how to fund most child and adolescent mental health services (CAMHS) is taken by National Health Service (NHS) Clinical Commissioning Groups (CCGs). The level of spend is not predetermined. CCGs are assigned an overall budget based on the physical and mental healthcare needs of the local population and commission services to meet those needs. As NHS England states, 'It is for CCGs to decide their priorities for spending.'[7] Analysis by the Royal College of Psychiatrists discovered a stark variation in spend on CAMHS contributing to a 'postcode lottery' in access to services.[8]

Variation in spend on CAMHS could be justified if it reflected real differences in need for mental health services among young people, but research suggests that it is only

BMJ

weakly associated with deprivation—a proxy for need—with 'stark unexplained variation'.[8] Indeed, recent evidence from England suggests that the budget allocation to CAMHS does not reflect levels of psychopathology. For instance, between 2012/2013 and 2014/2015 there was a sharp rise in referrals to CAMHS alongside a growing number of young people presenting at emergency departments with self-harm,[9 10] but some local CAMHS budgets were reduced in real terms over the same period.

Instead, it could be that spend on CAMHS reflects spending decisions for other services, either adult mental health or general and acute health services. The aim of this study was to assess the extent of the trade-off that exists between spend on CAMHS and other services. In order to quantify this we analysed the spending decisions of CCGs in England. Amid concerns that CAMHS are under-resourced and that recently earmarked additional national funding for CAMHS may be diverted to other areas,[11] it is important to better understand the funding allocation decisions taken by the commissioners.

## METHODS

### Data sources

We accessed publicly available data from a range of source. Data on spend for child and young person mental health (CYP MH) and adult mental health by CCG (n=209) were available from the Five Year Forward View Mental Health Dashboard (FYFVMHD).[12] NHS England report CYP MH spend. Although CCGs spend variable amounts on other services for CYP MH, such as acute paediatric care, in this paper we take overall spend as a proxy for spend on local CAMHS. Annual spend figures were available for the financial years 2016/2017 and 2017/2018. CCG population data were sourced from the Office of National Statistics (ONS).

Other data on CCG areas were identified from a range of publicly available UK data sets, including Public Health Fingertips, the NHS Outcomes Tool, NHS RightCare packs and Local Health. As far as possible data were used for the same year as CYP MH spend (2016/2017). Where this was not available, the nearest possible date was used; 2016 was chosen where data were available for calendar years. When data were reported for local authorities rather than CCGs, approximate values for CCGs were calculated using a 'lookup' file from the Local Government Association, which provides a link between local authority and CCG data. A full list of accessed data with time periods and sources is provided in the online supplementary file (online supplementary table S1).

### Spend on CYP MH

CCG reporting on CYP MH spend was introduced as part of the FYFVMHD. This figure covers the majority of CAMHS spend on tiers 2 (early help and targeted services) and 3 (specialised CAMHS services).[13] Spend on specialist eating disorder services is reported separately. CYP MH spend does not include inpatient services,

which are commissioned directly by NHS England and therefore not the subject of local allocation decisions, nor other services such as some children's psychological medicine or liaison services which are directly commissioned by acute healthcare trusts.

We used 2016/2017 spend in our analysis given the availability of data in other categories of our analysis. Spend was analysed in absolute terms and as a rate per young person aged 0–18, using CCG Mid-Year Population Estimates from the ONS.

### Potential explanatory factors of CAMHS spend variation

We selected a comprehensive set of indicators using a mix of evidence and consultation with health economic and mental health experts.

#### Category A: demand for CAMHS

We included 2015 local prevalence estimates for mental health conditions as an indicator of demand for CAMHS. Inpatient admissions to psychiatric units among those aged under 18 and rates of self-harm were also included as proxies of need. Other potential explanatory factors included the rate of children 'in need',[14] the rate of children in care, as well as a range of broader measures of vulnerability among young people as detailed in box 1.

#### Category B: demand for adult mental health services

For adult mental health, we included 2016/2017 mental health spend (excluding spend on children and young people and including primary care spend on mental health) from the FYFVMHD, 2015/2016 mental health primary care expenditure from NHS RightCare 'Where to Look' packs and, finally, the Quality and Outcome Frameworks (QOF) prevalence estimates for Depression and Severe Mental Illness from 2015/2016.

#### Category C: demand for physical health services

The need for physical health spend was proxied by a combination of spend, hospital admissions and prevalence rates. We included a range of physical health spend figures from 2015/2016 as reported in NHS RightCare 'Where to Look' packs. In the packs, spend was reported based on major International Classification of Disease-10 codes, further broken down into elective, non-elective and primary care spend and expressed as standardised rates. To calculate the crude rate of total physical health spend, we summed the numerators (absolute spend) across all physical health categories and expressed this as a rate using the respective CCG population. For the more detailed model, we summed the numerators for elective, non-elective and primary care spend for each condition and expressed this as a rate using the CCG population. We also included rates of elective and emergency hospital admissions. In addition, QOF prevalence estimates from 2015 to 2016 were used as alternative indicators of demand for physical healthcare. We chose prevalence estimates of this year because they were available at the point of the commissioning decisions in 2016/2017

## Box 1  Summary of potential explanatory factors of child and adolescent mental health services (CAMHS) spend variation

### Category A: demand for CAMHS
► Prevalence estimates (% aged 5–16), of: any mental health condition, conduct disorder, emotional disorders, hyperkinetic disorders.
► Child development at age 5 (%).
► General Certificate of Secondary Education (GCSE) achievement (%)—attaining 5 A–Cs including English and Maths.
► Child obesity (%), at: reception year, year 6.
► Children overweight (%), at: reception year, year 6.
► Children in need (rate per population aged 0–18): episodes per year, children per year, rate where mental health is a factor, rate where domestic violence is a factor, rate where cause is abuse or neglect, rate where cause is family stress.
► Special educational needs (rate per population aged 0–18).
► Hospital admissions for mental health under 18 (rate per person aged 0–18).
► Hospital admissions as a result of self-harm, population aged 10–24 (directly standardised rate).
► Children in care (rate per population aged 0–18): children in care, children leaving care looked-after-children where there is a cause for concern (rate per person aged 0–18).
► Children under 16 in poverty (%).
► Family homelessness (%).
► Not in education, employment or training (%).
► Rate of first time entrants to the youth justice system.

### Category B: demand for adult mental health services
► Quality and outcomes framework (QOF) prevalence estimates (%): depression, severe mental illness.
► Adult mental health spend (excluding CYP MH, but including learning disabilities).
► Mental health total primary care expenditure (£000).

### Category C: demand for physical health services
► All physical health spend per person (2015/2016).
► Hospital admissions: elective and emergency hospital admissions for all causes.
► Physical health spend per person (2015/2016), by condition: asthma, cancer (bowel, lung, reast, other), circulatory conditions (cerebrovascular disease, coronary heart disease (CHD), other), endocrine (diabetes, other), gastrointestinal (GI) (liver, lower GI, upper GI, other), genitourinary (renal, other), maternity, musculoskeletal, neurological, osteoarthritis, osteoporosis, respiratory (asthma, obstructive, other) admissions relating to fractures where a fall occurred, trauma and injuries.
► QOF prevalence estimates (%) for: asthma, cancer, chronic kidney disease, chronic obstructive pulmonary disease, diabetes, dementia, epilepsy, learning disabilities, heart failure, hypertension, obesity, osteoporosis, rheumatoid arthritis, palliative care, peripheral arterial disease, CHD, stroke and transient ischaemic attack and atrial fibrillation.

### Category D: general factors
► Index of Multiple Deprivation.
► Unemployment: unemployment (%), long-term unemployment (%)
► Ethnicity: BME (%), population whose ethnicity is not 'white UK' (%).
► Total CCG core allocation per capita.
► Distance from target allocation.
► Market forces factor.

Continued

## Box 1  Continued

► Population aged 65+ (%), population aged 85+ (%) overcrowding.
► Hospital stays for alcohol-related harm.

BME, black and minority ethnic; CCG, Clinical Commissioning Groups; CYP MH, child and young person mental health.

and were expected to be associated with the variable of interest.

### Category D: general factors
More general factors associated with health needs and spend on CYP MH were also included in the analysis. This included the level of deprivation, as measured by the Index of Multiple Deprivation, unemployment and long-term unemployment. Research also suggests that ethnic minority status may be linked to unmet need—the Children's Commissioner found children of Asian background were under-represented in CAMHS[15]—therefore the proportion of the population from black and minority ethnic (BME) groups was considered. The core allocation per person for each CCG was included to control for differences in budget. We included the difference between what NHS England's allocation formula suggests an area should get (target allocation) and the actual amount allocated in a given year—the 'distance from target allocation'. The two values diverge because of rules limiting the pace of change, that is, the amount by which allocations can change year-on-year. We also included the Market Forces Factor, an index that adjusts for variation in costs, as well as the per cent of the population aged 65 years or more and 85 years or more. Finally, we included the rates of overcrowding of households and hospital stays for alcohol-related harm.

### Statistical analysis
To explain variation in spend on CYP MH across CCGs, we reviewed the proportion of CCG budgets spent on CYP MH. We next inspected the association between the proportion spent on CYP MH and the total CCG budget, as well as the association between CYP MH spend per CCG and the population of the CCG aged 0–18.

To investigate statistical associations between CYP MH spend per person aged 0–18 and the explanatory factors identified, we followed a two-step analytical process. First, we selected factors from each category listed in table 1 that were associated with the dependent variable (ie, spend on CYP MH per person aged 0–18). To do this, we included all factors in each category in a generalised linear model (GLM) and used likelihood ratio tests to select backwards those factors that gave the best goodness-of-fit of the model to the data—an approach known as 'Block-Wise Selection'.[16] The order in which the factors were tested by likelihood ratio tests was determined by the p value from the regression (from largest to smallest p value).

Second, all selected factors from each category were included in a GLM and the regression coefficients were

**Table 1** Association of CYP MH spend per person aged 0–18 with explanatory factors (main model)

| | Main analysis | SA 1—excluding outliers | SA 2—including eating disorders |
|---|---|---|---|
| | Ratio of means (SE) (95% CI) | Ratio of means (SE) (95% CI) | Ratio of means (SE) (95% CI) |
| Children entering youth justice per 1000 aged 0–18 | 1.04 (0.03) (0.99 to 1.09) | 1.02 (0.02) (0.97 to 1.07) | 1.03 (0.03) (0.98 to 1.09) |
| Children in need (family stress), rate per 100 aged 0–18 | 1.05 (0.05) (0.96 to 1.15) | 1.04 (0.04) (0.97 to 1.13) | 1.06 (0.05) (0.98 to 1.16) |
| Obese children (reception year) | 0.94 (0.05) (0.84 to 1.05) | 0.95 (0.05) (0.86 to 1.05) | 0.94 (0.05) (0.84 to 1.05) |
| Children with excess weight (reception year) | 1.05 (0.04) (0.98 to 1.12) | 1.05 (0.03) (0.99 to 1.11) | 1.04 (0.03) (0.98 to 1.11) |
| Children in care per 1000 aged 0–18 | 1.04* (0.02) (1.00 to 1.07) | 1.03* (0.01) (1.00 to 1.05) | 1.04* (0.02) (1.01 to 1.07) |
| Spend on MH per person over 18 (excl. CYP, £10) | 1.01 (0.01) (1.00 to 1.03) | 1.02** (0.01) (1.00 to 1.03) | 1.01* (0.01) (1.00 to 1.03) |
| All physical health spend per person (£100) | 0.91** (0.04) (0.84 to 0.99) | 0.93* (0.03) (0.87 to 1.00) | 0.92** (0.04) (0.85 to 1.00) |
| Percentage of black and minority ethnic | 0.99** (0.00) (0.98 to 1.00) | 0.99* (0.00) (0.98 to 1.00) | 0.99* (0.00) (0.99 to 1.00) |
| Overcrowding | 1.01 (0.01) (0.99 to 1.03) | 1.01 (0.01) (0.99 to 1.02) | 1.01 (0.01) (0.99 to 1.02) |
| Long-term unemployment | 1.03 (0.02) (1.00 to 1.06) | 1.02 (0.01) (0.99 to 1.04) | 1.02 (0.01) (0.99 to 1.05) |
| Observations | 199 | 190 | 199 |
| Akaike information criterion (AIC) | 1935 | 1855 | 1963 |
| Bayesian information criterion (BIC) | 1971 | 1890 | 2000 |
| Pseudo R-squared | 0.29 | 0.31 | 0.28 |

*P<0.05, **P<0.01, ***P<0.001.
CYP MH, child and young person mental health; SA, sensitivity analysis.

presented in the exponentiated form to give the ratio of means (ie, the proportional adjustment in the dependent variable 'CYP MH spend per person aged 0–18' expected from a one unit increase in the covariate). All GLMs were specified with gamma distribution and log link function because this combination provided the best goodness-of-fit based on a modified Park test (using the user-command 'GLMdiag' in STATA) and the AIC and BIC information criteria. We ran the analysis in the main model featuring the rate of all physical health spend. To understand the relationship in detail, we also ran a second, detailed physical care model, which included spend broken down by category.

### Sensitivity analyses
To address the uncertainty in the results, we conducted three sensitivity analyses. First, we removed CCGs with extreme low and high spend on CAMHS from the analysis to address concerns that CAMHS spend figures are incorrectly recorded for some CCGs.[17] The extreme CAMHS spend values (or outliers) were identified using studentised deviance residuals. Second, we added spend on eating disorders in the dependent variable (ie, spend on CYP MH per person aged 0–18) because some CCGs have been unable to split out these costs due to block contracting and it was unclear if, for some CCGs, these costs were excluded entirely from the dependent variable.[11] Third, QOF prevalence estimates were used as indicators of physical health need instead of spend on physical care because physical health spend data were out of date by a year compared with the CAMHS spend data.

### Patient and public involvement
This research was done without patient involvement. Patients were not invited to comment on the study design and were not consulted to develop patient-relevant outcomes or interpret the results. Patients were not invited to contribute to the writing or editing of this document for readability or accuracy.

### RESULTS
Spend on CYP MH (excluding spend on specialist eating disorder services) accounted on average for 0.8% of CCGs' budget in 2016/2017. As shown in figure 1, it appears that the proportion of CCG budget spent on CYP MH has a weak positive association with the size of the CCG budget and a strong positive association with the size of the CYP population.

The average spend on CYP MH per person aged 0–18 in 2016/2017 across 209 CCGs was £46 (SD=£18, min=£2, max=£127) with considerable variation across CCGs in England (figure 2).

The results from the main regression analysis are presented in table 1 and show that an additional percentage point in the proportion of children in care is associated with 4% higher CYP MH spend per person aged 0–18 (ratio of means: 1.04; 95% CI 1.00 to 1.07). By contrast, spending £100 more on physical health services was associated with 9% lower spend in CYP MH per person aged 0–18 (ratio of means: 0.91; 95% CI 0.84 to 0.99). The proportion of the population of BME also appeared to be

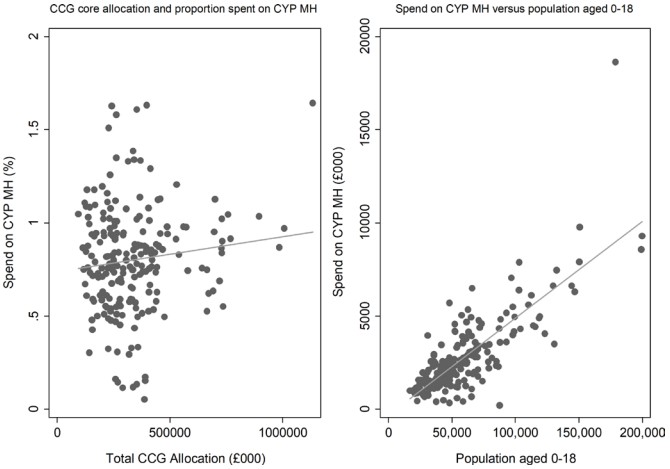

**Figure 1** Scatter plots: spend on child and young person mental health services. CCG, Clinical Commissioning Groups; CYP MH, child and young person mental health.

negatively associated with the dependent variable but the magnitude was small (ratio of means: 0.99; 95% CI 0.98 to 1.00). Overall, the main regression model explained 29% of the variation in CYP MH spend per person. The summary statistics for the explanatory factors included in the main regression model are presented in the online supplementary file table 1.

Regarding the first sensitivity analysis, seven CCGs were excluded because they spent less than £11 per person aged 0–18 on CYP MH and two CCGs were excluded because they spent above £117 per head. The results of this sensitivity analysis as well as the sensitivity analysis where spend on eating disorders was added in the dependent variable were very similar to the results of the main analysis. The most noticeable difference was that spending £10 more on adult mental health was significantly and positively associated with 2% (ratio of means: 1.02; 95% CI 1.00 to 1.03) and 1% (ratio of means: 1.01; 95% CI 1.00 to 1.03) more spend on CYP MH spend per person aged 0–18, respectively. Also, the negative association between

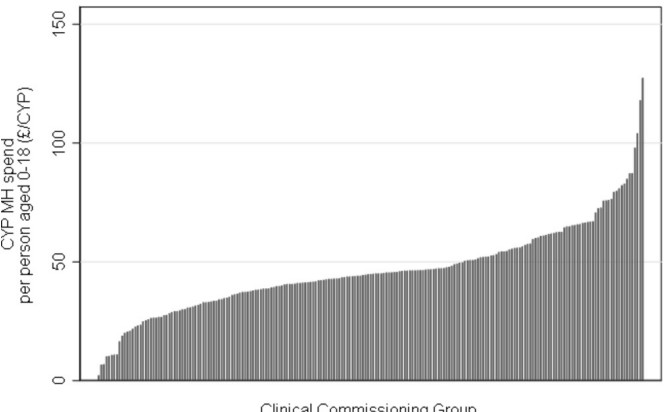

**Figure 2** Spend on child and young person mental health services, by CCG. CCG, Clinical Commissioning Groups; CYP MH, child and young person mental health.

spend on physical health with the dependent variable was reduced from 9% in the main analysis to 7% (ratio of means: 0.93; 95% CI 0.87 to 1.00) and 8% (ratio of means: 0.92; 95% CI 0.86 to 0.98), respectively.

The results from the detailed physical care regression model are presented in table 2 and show that only spend on circulatory conditions (0.98; 95% CI 0.96 to 0.99) and the proportion BME (0.99; 95% CI 0.98 to 1.00) were associated with the rate of spend on CYP MH. Specifically, an additional pound spent on 'other circulatory conditions' per person was associated with a 2% decrease in spend per head on CYP MH. When including QOF prevalence estimates in the detailed model instead of spend on physical health (third sensitivity analysis), there were no significant associations, although the point estimates for cancer, diabetes and chronic obstructive pulmonary disease were each negative.

## DISCUSSION

Healthcare commissioners in England face a challenge in balancing competing needs. This paper contributes to our understanding of this by quantifying the bilateral associations, or trade-offs, with CYP MH spend when allocating budgets. Our main findings are that per capita spend on CYP MH was positively associated with need, the level of children in need stands out as significant among the variables selected; positively associated with spend on adult mental health; but negatively associated with spend on physical health and the proportion of the population which is from BME backgrounds.

In principle, with homogeneous preferences and perfect information on both the needs and optimal treatments to inform the allocation of budgets, we would not expect a trade-off with physical health to exist once accounting for variation in need and the size of the population. Variation in need alone would explain differences in spend on CYP MH. In practice, the decentralised system of healthcare commissioning in England offers a number of possible reasons for the identified association. For instance, the configuration of local services may differ; there may be variation in patient preferences; or commissioners might have different preferences when allocating funding.[18] Decisions are also constrained by history: resistance to large-scale disinvestment means that in any given year the scope for reallocating budgets across competing claims is likely to be marginal at the best.[19]

However, concerns persist that commissioners do not readily prioritise CAMHS. Indeed, CAMHS has been described as the 'Cinderella of the Cinderella service' in that despite repeated promises of additional funding it has arguably remained under-resourced relative to levels of need.[9] Although the associations identified as part of this paper can do nothing to prove or disprove this, claims that national funding earmarked for CAMHS may have been diverted to other services raises the question as to whether incentives or biases exist that give priority to physical health.[11] We explore possible reasons below.

**Table 2** Association of CYP MH spend per person aged 0–18 with explanatory factors (detailed physical care model)

| | Detailed spend | SA 3 – QOF prevalence |
|---|---|---|
| | Ratio of means (SE) (95% CI) | Ratio of means (SE) (95% CI) |
| Children entering youth justice per 1000 aged 0–18 | 1.03 (0.03) (0.97 to 1.09) | 1.04 (0.03) (0.99 to 1.10) |
| Children in need (family stress), rate per 100 aged 0–18 | 1.07 (0.05) (0.98 to 1.18) | 1.07 (0.05) (0.97 to 1.17) |
| Obese children (reception year) | 0.95 (0.06) (0.84 to 1.07) | 0.96 (0.06) (0.85 to 1.08) |
| Children with excess weight (reception year) | 1.04 (0.04) (0.97 to 1.12) | 1.04 (0.04) (0.98 to 1.12) |
| Children in care per 1000 aged 0–18 | 1.03 (0.02) (1.00 to 1.07) | 1.02 (0.02) (0.99 to 1.06) |
| Spend on MH per person over 18 (excl. CYP, £10) | 1.01 (0.01) (1.00 to 1.02) | 1.01 (0.01) (1.00 to 1.02) |
| Spend on circulatory disease per person: coronary heart disease | 1.01 (0.01) (0.98 to 1.03) | |
| Spend on circulatory disease per person: other | 0.98** (0.01) (0.97 to 0.99) | |
| Spend on respiratory disease per person: obstructive conditions | 1.01 (0.02) (0.98 to 1.04) | |
| Spend on cancer per person: bowel cancer | 1.02 (0.04) (0.95 to 1.10) | |
| Spend on maternity per person | 1.18 (0.10) (0.99 to 1.40) | |
| Percentage of black and minority ethnic | 0.99* (0.00) (0.98 to 1.00) | 0.99 (0.01) (0.98 to 1.01) |
| Overcrowding | 1.01 (0.01) (0.99 to 1.03) | 1.01 (0.01) (0.99 to 1.02) |
| Long-term unemployment | 1.02 (0.02) (0.99 to 1.05) | 1.03 (0.02) (1.00 to 1.06) |
| Cancer prevalence—18+, QOF | | 0.93 (0.09) (0.77 to 1.13) |
| Diabetes prevalence—18+, QOF | | 0.94 (0.03) (0.86 to 1.03) |
| COPD prevalence, QOF | | 0.98 (0.09) (0.82 to 1.18) |
| Observations | 199 | 199 |
| Akaike information criterion (AIC) | 1942 | 1939 |
| Bayesian information criterion (BIC) | 1991 | 1981 |
| Pseudo R-squared | 0.32 | 0.29 |

**P<0.05, **P<0.01, ***P<0.001.
COPD, chronic obstructive pulmonary disease; CYP MH, child and young person mental health; QOF, Quality and Outcome Frameworks; SA, sensitivity analysis.

First, spending decisions may be biassed towards a 'rule of rescue'. This predicts that spend will gravitate towards cases of immediate and pressing need and away from what might be considered preventative.[20] It may be that CYP MH is perceived as an area where rationing by deterrence or delay is more acceptable than in other areas, especially as some of the benefits are long term. Indeed, long waits for CAMHS appear to evidence this.[11]

Second, the lower quality and visibility of data relating to CYP MH compared with care for adults may have influenced spending decisions.[21] Historically, there has been more systematic and thorough recording and measurement of processes relating to acute health conditions such as A&E waiting times and hospital length of stay. By contrast, until recently there was no structured reporting of national data on processes and outcomes in CAMHS, which may have influenced commissioners' allocation decisions.

Third, the general level of awareness or stigma around mental health may also contribute to low prioritisation of CYP MH.[22] The positive association between spend on adult and CYP mental health found in this paper could be explained by commissioners with increased awareness or knowledge of mental health spending more on mental health services for both CYP and adults.[23] Furthermore, the negative association of spend on CYP MH with the proportion of the BME population found in this study may reflect lower levels of awareness of mental health in BME communities.[24]

Finally, as has been argued elsewhere, financial incentives exist that encourage the adoption of new technologies in healthcare.[25] New technologies are one of the main drivers of spending in the NHS.[26] The lobbying for and adoption of new technologies may favour interventions like innovations in surgical instruments and certain drugs (eg, those used in oncology) over interventions which are primarily labour-intensive, such as those in CAMHS.

We have been concerned with trade-offs, but it would be misleading to posit physical and mental health purely as substitutes when there are complementarities and services may benefit from being more closely integrated. Good mental health can support physical health and vice versa.[27] Also, more efficient services anywhere in the system could help free resources. For instance, in

our detailed spend model, we found that spend on CYP MH was negatively associated with spend on some circulatory conditions, one of the highest spend disease groups in the UK.[28] Better prevention and early intervention in the community for these diseases may release resources to help meet the increasing demand for mental health services and in particular CAMHS.

This research has highlighted the extent of the trade-off between physical health spend and spend on CYP MH. In light of concerns that CAMHS is not given the priority it merits we have also suggested factors that could bias spend towards physical health. We would recommend addressing the inequality in the availability and quality of data for CYP MH to help minimise, or at least make more explicit, unwarranted variation in spend on CAMHS. Steps are already being taken: recently published estimates of the prevalence of mental health disorders among young people may better capture variation in need[29]; CYP MH spend is being monitored as part of the FYFVMH dashboard; and new targets are being introduced, such as for access rates and waiting times, to help commissioners understand how well services are performing.[12] Future research should focus on the outcomes arising from different levels of investment in CAMHS and other services that support young people.

### Strengths and limitations

The main limitation of this paper is the availability of data. Limited to a single year of spending, this paper can provide evidence only of association and not causality. There was also a lack of contemporaneous and consistent data. In particular, spend on physical health conditions was from 2015/2016, although the distribution of spend across different programmes is unlikely to have altered dramatically. Other indicators are not available at CCG level but are estimated from local authority data, specifically this includes a range of factors explaining demand for CYP MH services. Finally, as highlighted above, there are concerns over the consistency and accuracy of the figures published for spend on CYP MH, with challenges in breaking out different facets of expenditure. This is addressed by the sensitivity analyses included.

The major strength of this study is how we have been able to use and combine different area-level data from a range of datasets, including efforts to incorporate local authority data to understand factors driving need for CYP MH services. Recent research into variation in Medicare spend in the USA found that area-level factors were the most appropriate way to analyse such variation.[30] Seeking a parsimonious model amid the array of available data, we applied a stepwise selection procedure. The main criticism of this set of procedures is that they are too data driven. In order to overcome this, we took a block-wise approach.[16] This meant we retained the theoretical foundation outlined, while still being data driven in our attempts to limit the number of independent variables included. The incorporation of sensitivity analyses

to address uncertainty in the results also strengthen this study.

### CONCLUSIONS

Healthcare commissioners in England face a challenge in balancing competing needs. This paper contributes to our understanding of this by quantifying the bilateral associations with CYP MH spend when allocating budgets. Any attempt to explain the variation in CAMHS spend must also take account of demand for other services.

**Contributors** SR carried out the data acquisition, conducted the analysis and drafted the manuscript. AT also contributed to the study design, analysis plan and helped refine the manuscript. MF provided input into the study design and provided feedback on the manuscript. All authors read and approved the final version.

**Funding** This research was funded by the National Institute for Health Research (NIHR) Collaboration for Leadership in Applied Health Research and Care Oxford at Oxford Health NHS Foundation Trust.

**Disclaimer** The views expressed are those of the author(s) and not necessarily those of the NHS, the NIHR or the Department of Health and Social Care.

**Competing interests** None declared.

**Patient consent for publication** Not required.

**Ethics approval** This study uses publicly available data only and therefore further ethical approval was not required.

**Provenance and peer review** Not commissioned; externally peer reviewed.

**Data availability statement** All data relevant to the study are included in the article or uploaded as online supplementary information.

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
