## [Reviewer comments · BMJ Open]

ARTICLE DETAILS

TITLE (PROVISIONAL)	Variation in spend on young mental health across Clinical Commissioning Groups in England: a cross-sectional observational study
AUTHORS	Rocks, Stephen; Fazel, Mina; Tsiachristas, Apostolos

VERSION 1 - REVIEW

REVIEWER	Ingrid Zechmeister-Koss Ludwig Boltzmann Institute for HTA Austria
REVIEW RETURNED	08-May-2019

GENERAL COMMENTS	The paper addresses a significant question of resource allocation in the health care system. This is a very relevant topic given the situation that very expensive health technologies (mainly drugs and medical devices) have recently entered the health care market and demand a very high proportion of spending while areas which are less 'technology-driven' such as mental health care do not have the same strong lobbies (power) to "sell" treatments to payers. The authors' hypothesis is therefore worthwhile to test. Method 1) I missed a statement on the significance level applied and how the multiple comparison situation was handled.2) I missed a more substantial justification for the selection of the backward-elimination approach and some notes on its limitations in the discussion section. For example, the backward-elimination approach is less theory-driven and more data-driven regarding the explanatory variables that are finally selected via the automatic algorithm applied in the approach.3) I was not able to read figure 2 – wouldn't a simple histogram do the job?4) I am not able to comment on the multiple data sources that were used to populate the analyses because I am not familiar with the characteristics of UK administrative data. I am concerned, however, by the fact that the analysis does not include data on inpatient care in CAMH which – at least in my country – would represent quite a significant proportion of the total spending. Interpretation of results I would like to challenge some of the interpretations of the results. 1) p.10/line 39: I am not convinced about the first interpretation "Our findings indicate that healthcare commissioners in England face a challenge in balancing competing needs and there appears
---

	to be a trade-off physical health and CYP MH when allocating budgets” – asked provocatively: How would the results to have look like to prove the opposite? 2) p. 10/line 42: I am unsure if I understood this interpretation correctly: "This trade-off persists when local needs are taken into account, implying something beyond the variation expected under a devolved healthcare commissioning structure." Do you imply here that the differences in expenditure can't be fully explained by your explanatory variables and that this means that the differences are not justified? This in turn would mean that you were able to capture all explanatory variables in your analysis which in my view is questionable. If further explanatory variables exist the result may change as you showed in your sensitivity analyses. 3) p. 10/line 48: My other comment is related to your argument "The second is that CYP MH does not appear to be readily prioritised by commissioners." If expenditure correlate negatively one can imply that for all other areas as well. So in turn, where there is more spending on CYP MH there is less for other areas. Depending on the perspective, one could then also say that physical health is not readily prioritised by commissioners. My argument here is that the interpretation based on proportions of spending is not convincing to me because in my view it cannot be concluded if it is too little, enough or if there is overspending – this would require a link to efficiency and thus to the effectiveness of services and interventions across different areas. 4) One could also question whether demand is per se a valid explanatory concept for appropriate health care spending. As we all know it is influenced by many factors and high demand does not automatically justify high spending. This is particularly complicated in mental health because here we are dealing with the fact that the demand is generally lower in CAMH because many people may not actively seek professional help – e.g. figures from an Austrian survey have shown that less 50 % of adolescents with diagnosed mental health problems do not seek help. So the variable "inpatient admissions to psychiatric units for <18years old" may be already biased in itself as an indicator for demand. On the other hand, if the argument holds that CAMH is underfunded then the use of hospital care is in itself already influenced by limited available capacities and does not reflect demand either. I also missed some arguments in the discussion that address the issue of power imbalance and hierarchies between medical areas (e.g. psychiatry vs. surgery) and the lack of industry that pushes spending on expensive drugs and technologies in other areas. (e.g. orphan drugs, robotic surgery, oncology). 5) The paper is based on the notion of a competition between physical and mental health. Is it a good way forward to reinforce the 'competition paradigm' between physical and mental health?
--	--

REVIEWER	Yan Feng Queen Mary University of London, UK
REVIEW RETURNED	13-May-2019

GENERAL COMMENTS	This paper explores whether the rate of spend on child and adolescent mental health (MH) is influenced by demand for other
--

competing services in local commissioning decisions. The study is conducted based on data from 209 Clinical Commission Groups (CCGs) in England. The results suggest that the child and adolescent mental health is poorly prioritised when compared with physical health in local budget decision.

I really enjoyed reading this paper. Not only because it tries to answer an important research question but also the elegant way that the authors address the practical (and messy) problems with the data collection. Furthermore, the research work is very well presented.

I have a number of comments mainly on the specification of the statistical analysis. Hope the authors might find it useful.

(1) The main analysis is about modelling expenditure in child and adolescent MH problems at CCG level as a function of expenditure in adults MH problems, expenditure in physical problems, and other items. My concern is that the three types of expenditures (MH problems for child and adolescents, MH problems for adults, and physical problems) at CCG level are jointly determined at the same time. In another word, the only true exogenous variable in the main model is budget at CCG level. As authors wrote in the second paragraph of the paper "CCGs are assigned an overall budget based on need to commission a range of health services for their local population, but as NHS England states, 'it is for CCGs to decide their priorities for spending'". The current specification might lead to biased estimates.

(2) Instead of including expenditures from adults MH expenditure and physical health expenditure, the really driven force behind each type of expenditure is local populations' needs. When CCG decision makers consider of allocating health care budgets, they should look at the need of its local population between various diseases. To reflect this decision making process, instead of having two types of expenditures in the model as independent variables, authors might consider of including two sets of needs variables.

(3) When the budget for CCG is "fixed", it is not a surprise that an increase in the expenditure of one area might lead to a drop/increase in another area. The authors contributed very important evidence in this research area. However, I consider the fundamental question to address is that whether the budget allocation truly reflect the local population's health need. To achieve this, the authors might estimate three expenditure equations, with each type expenditure as an equation. My suggested specification in point (2) should be able to address this.

(4) My final comments on the modelling part is about the "other variables". The authors successfully included variables like unemployment rate to control for the variance of social demographics between populations in different CCGs. However, some other environmental variables might be considered too, like price difference between CCGs (Martin et al., 2008).

(5) Based on my comments on the modelling specification, I do not think the authors are able to claim whether the current budget allocation can (or cannot) reflect the really health need for local population. In particular I refer to authors' statement in the last sentences of the first paragraph in the discussion section. "The first is that the budgets allocation to CCGs from NHS England do not seem to reflect the real need for physical and

	mental health services. The second is that CYP MH does not appear to be readily prioritised by commissioners”. (6) The modelling work is based on very small sample (under 200 data points) for one year only (FY 2016/17). Therefore, the results derived from this study cannot reflect accurate causality relationship but association relationship only. The authors might consider of making this point explicit to avoid confusions by readers with no strong statistical background. Reference S. Martin, N. Rice, P.C. Smith. Does health care spending improve health outcomes? Evidence from English programme budgeting data. J Health Econ, 27 (2008), pp. 826-842
--	---

VERSION 1 – AUTHOR RESPONSE

Reviewer 1

Comment: I missed a statement on the significance level applied and how the multiple comparison situation was handled.

DONE: The statistical level handled in the regression analyses was presented below Table 2 and Table 3 by using stars, i.e. * $p < 0.05$, ** $p < 0.01$, *** $p < 0.001$. The column headings in these tables also refer to the 95% CI.

Comment: I missed a more substantial justification for the selection of the backward-elimination approach and some notes on its limitations in the discussion section. For example, the backward-elimination approach is less theory-driven and more data-driven regarding the explanatory variables that are finally selected via the automatic algorithm applied in the approach.

CLARIFICATION PROVIDED: The approach we used - also referred to as “Block-Wise Selection” - 1 seeks to overcome the primary criticism of selection models as being too data driven. Specifically, by specifying the framework in advance and selecting variables within this, we retained our theoretical foundation but also used a data-driven approach to attain a parsimonious model. We have added the following to the discussion:

“Seeking a parsimonious model amid the array of available data, we applied a stepwise selection procedure. The main criticism of this set of procedures is that they are too data driven. In order to overcome this, we took a block-wise approach. This meant we retained the theoretical foundation outlined, while still being data driven in our attempts to limit the number of independent variables included.”

Comment: I was not able to read figure 2 – wouldn’t a simple histogram do the job?

DONE: Thank you. We agree and have amended the existing figure accordingly, including removing the national average and making the titles clearer.

Comment: I am not able to comment on the multiple data sources that were used to populate the analyses because I am not familiar with the characteristics of UK administrative data. I am concerned, however, by the fact that the analysis does not include data on inpatient care in CAMH which – at least in my country – would represent quite a significant proportion of the total spending.

CLARIFICATION PROVIDED: Inpatient care, while important, is commissioned nationally and therefore not subject to the type of local budget allocation decision which is the subject of the paper.

To make this clearer to international readers we revised the “Spend on child and young person mental health” section in the Methods, which now reads:

“CCG reporting on CYP MH spend was introduced as part of the FYFVMHD. This figure covers the majority of CAMHS spend on tiers 2 (early help and targeted services) and 3 (specialised CAMHS services). Spend on specialist eating disorder services is reported separately. CYP MH spend does not include inpatient services, which are commissioned directly by NHS England and therefore not the subject of local allocation decisions, nor other services directly commissioned by other sources such as some children’s psychological medicine or liaison services commissioned by acute healthcare trusts.”

Comment: p.10/line 39: I am not convinced about the first interpretation “Our findings indicate that healthcare commissioners in England face a challenge in balancing competing needs and there appears to be a trade-off physical health and CYP MH when allocating budgets” – asked provocatively: How would the results to have look like to prove the opposite?

CLARIFICATION PROVIDED: We revised the first paragraph in the Discussion to make clearer that the findings only show that a bilateral association, or trade-off, exists, rather than stating that this in itself shows that CAMHS is less of a priority. The revised paragraph now reads:

“Healthcare commissioners in England face a challenge in balancing competing needs. This paper contributes to our understanding of this by quantifying the bi-lateral associations, or trade-offs, with CYP MH spend when allocating budgets. Our main findings are that per capita spend on CYP MH was positively associated with need, the level of children in need stands out as significant among the variables selected; positively associated with spend on adult mental health; but negatively associated with spend on physical health and the proportion of the population which is from black and minority ethnic backgrounds.”

We also added a paragraph that explores specifically this issue of variation in spend. As explained in the Introduction, NHS England allocates healthcare budget to CCGs based on an algorithm that estimates separately the budget needed for mental and physical care services. On the basis of these estimates, and accounting for variation in need and population, we wouldn’t expect to find any association between the spend in mental and physical care. However, in light of the allocation decisions taken by healthcare commissioners we do observe an association, or trade-off. We have therefore added the following to our discussion:

“In principle, with homogenous preferences and perfect information on both the needs and optimal treatments to inform the allocation of budgets, we would not expect a trade-off with physical health to exist once accounting for variation in need and the size of the population. Variation in need alone would explain differences in spend on CYP MH. In practice, the decentralised system of healthcare commissioning in England offers a number of possible reasons for the identified association, for instance: the configuration of local services may differ; there may be variation in patient preferences; or commissioners might have different preferences when allocating funding. Decisions are also constrained by history: resistance to large-scale disinvestment means that in any given year the scope for reallocating budgets across competing claims is likely to be marginal at best.”

Comment: p. 10/line 42: I am unsure if I understood this interpretation correctly: "This trade-off persists when local needs are taken into account, implying something beyond the variation expected under a devolved healthcare commissioning structure." Do you imply here that the differences in expenditure can't be fully explained by your explanatory variables and that this means that the differences are not justified? This in turn would mean that you were able to capture all explanatory variables in your analysis which in my view is questionable. If further explanatory variables exist the result may change as you showed in your sensitivity analyses.

DONE: We agree this is a potentially confusing statement and have removed it from the text. The general point links to the above, that the variation in spend reflects more than just variation in need. However, this does reflect the devolved commissioning structure, where commissioners' preferences, among other factors, influence allocation decisions.

Comment: p. 10/line 48: My other comment is related to your argument "The second is that CYP MH does not appear to be readily prioritised by commissioners." If expenditure correlate negatively one can imply that for all other areas as well. So in turn, where there is more spending on CYP MH there is less for other areas. Depending on the perspective, one could then also say that physical health is not readily prioritised by commissioners.

DONE: We agree with this point, thank you. We have revised the text to be explicit that the associations found in this paper cannot prove or disprove the hypothesis that CAMHS is underfunded, but in light of the existence of a possible trade-off and with anecdotal evidence that funding is siphoned off from CAMHS to meet other needs, we go on to explore potential reasons for a bias. The specific sentence now reads:

"Although the associations identified as part of this paper can do nothing to prove or disprove this, claims that national funding earmarked for CAMHS may have been diverted to other services raises the question as to whether incentives or biases exist that may give priority to physical health."

Comment: My argument here is that the interpretation based on proportions of spending is not convincing to me because in my view it cannot be concluded if it is too little, enough or if there is overspending – this would require a link to efficiency and thus to the effectiveness of services and interventions across different areas.

CLARIFICATION PROVIDED: As above, we agree with this point. We have revised the first paragraph in the Discussion to make clearer that the findings only show that a bilateral association, or trade-off, exists, rather than stating that this in itself shows that CAMHS is less of a priority. The revised paragraph now reads:

"Healthcare commissioners in England face a challenge in balancing competing needs. This paper contributes to our understanding of this by quantifying the bi-lateral associations, or trade-offs, with CYP MH spend when allocating budgets. Our main findings are that per capita spend on CYP MH was positively associated with need, the level of children in need stands out as significant among the variables selected; positively associated with spend on adult mental health; but negatively associated with spend on physical health and the proportion of the population which is from black and minority ethnic backgrounds."

Comment: One could also question whether demand is per se a valid explanatory concept for appropriate health care spending. As we all know it is influenced by many factors and high demand does not automatically justify high spending. This is particularly complicated in mental health because here we are dealing with the fact that the demand is generally lower in CAMH because many people may not actively seek professional help – e.g. figures from an Austrian survey have shown that less 50 % of adolescents with diagnosed mental health problems do not seek help. So the variable "inpatient admissions to psychiatric units for <18years old" may be already biased in itself as an indicator for demand. On the other hand, if the argument holds that CAMH is underfunded then the use of hospital care is in itself already influenced by limited available capacities and does not reflect demand either.

CLARIFICATION PROVIDED: We have added a paragraph in the beginning of the Discussion section to reflect the range of factors that could account for the variation in spend. the paragraph, as also described above, now reads:

“In principle, with homogenous preferences and perfect information on both the needs and optimal treatments to inform the allocation of budgets, we would not expect a trade-off with physical health to exist once accounting for variation in need and the size of the population. Variation in need alone would explain differences in spend on CYP MH. In practice, the decentralised system of healthcare commissioning in England offers a number of possible reasons for the identified association, for instance: the configuration of local services may differ; there may be variation in patient preferences; or commissioners might have different preferences when allocating funding. Decisions are also constrained by history: resistance to large-scale disinvestment means that in any given year the scope for reallocating budgets across competing claims is likely to be marginal at best.”

Comment: I also missed some arguments in the discussion that address the issue of power imbalance and hierarchies between medical areas (e.g. psychiatry vs. surgery) and the lack of industry that pushes spending on expensive drugs and technologies in other areas. (e.g. orphan drugs, robotic surgery, oncology).

DONE: This is an interesting argument that not only complements, but builds on some of the points we have made already. We have added a further point that briefly broaches this – as this in itself could be the subject of a far more lengthy discussion. We added this argument in a new a paragraph in the Discussion that reads:

“Finally, as has been argued elsewhere, financial incentives exist that encourage the adoption of new technologies in healthcare. New technologies are one of the main drivers of spending in the NHS. The lobbying for and adoption of new technologies may favour interventions like innovations in surgical instruments and certain drugs (e.g. those used in oncology) over interventions which are primarily labour-intensive such as those in CAMHS.”

Comment: The paper is based on the notion of a competition between physical and mental health. Is it a good way forward to reinforce the ‘competition paradigm’ between physical and mental health?

CLARIFICATION PROVIDED: We fully agree with the reviewer that it is important to promote an approach which integrates physical and mental health, however the present structure of services reinforces the current paradigm and so our results reflect current configuration of services. However, we agree that this is important and so have added a new paragraph in the Discussion that reads:

“We have been concerned with trade-offs, but it would be misleading to posit physical and mental health purely as substitutes when there are complementarities and services may benefit from being more closely integrated. Good mental health can support physical health and vice versa. Also, more efficient services anywhere in the system could help free resources. For instance, in our detailed spend model we found that spend on CYP MH was negatively associated with spend on some circulatory conditions, one of the highest spend disease groups in the UK. Better prevention and early intervention in the community for these diseases may release resources to help meet the increasing demand for mental health services and in particular CAMHS.”

Reviewer 2

Comment: The main analysis is about modelling expenditure in child and adolescent MH problems at CCG level as a function of expenditure in adults MH problems, expenditure in physical problems, and other items. My concern is that the three types of expenditures (MH problems for child and adolescents, MH problems for adults, and physical problems) at CCG level are joined determined at the same time. In another word, the only true exogenous variable in the main model is budget at CCG level. As authors wrote in the second paragraph of the paper “CCGs are assigned an overall budget based on need to commission a range of health services for their local population, but as NHS

England states, 'it is for CCGs to decide their priorities for spending'. The current specification might lead to biased estimates.

CLARIFICATION: We agree which is why in our study we have restricted our findings to the direction of the association, not any causal relation which would be subject to omitted variable bias.

Comment: Instead of including expenditures from adults MH expenditure and physical health expenditure, the really driven force behind each type of expenditure is local populations' needs. When CCG decision makers consider of allocating health care budgets, they should look at the need of its local population between various diseases. To reflect this decision making process, instead of having two types of expenditures in the model as independent variables, authors might consider of including two sets of needs variables.

CLARIFICATION: This is a fair point. Although in our main model we were interested in the overall category of physical health, we investigated the relationship with need (as defined by prevalence) instead of spend in a sensitivity analysis. While none of the prevalence estimates were significant, each had a negative association – the same direction as overall physical spend – which, we believe, lends support to the argument of a trade-off between different needs.

Comment: When the budget for CCG is "fixed", it is not a surprise that an increase in the expenditure of one area might lead to a drop/increase in another area. The authors contributed very important evidence in this research area. However, I consider the fundamental question to address is that whether the budget allocation truly reflect the location population's health need. To achieve this, the authors might estimate three expenditure equations, with each type expenditure as an equation. My suggested specification in point (2) should be able to address this.

CLARIFICATION: As explained in the Introduction, NHS England allocates healthcare budget to CCGs based on an algorithm that estimates separately the budget needed for mental and physical care services. Thus, if these estimates were correct, we would not expect to find any association between the spend in mental and physical care. As our study found such an association it raises concerns about the budget allocation system among CCGs in England.

Comment: Based on my comments on the modelling specification, I do not think the authors are able to claim whether the current budget allocation can (or cannot) reflect the really health need for local population. In particular I refer to authors' statement in the last sentences of the first paragraph in the discussion section. "The first is that the budgets allocation to CCGs from NHS England do not seem to reflect the real need for physical and mental health services. The second is that CYP MH does not appear to be readily prioritised by commissioners".

CLARIFICATION PROVIDED: We fully agree with the reviewer and have revised the first paragraph in the Discussion section to make clear that the paper only shows that a bilateral association, or trade-off, exists, rather than stating that this in itself shows that CAMHS is less of a priority. The revised paragraph now reads:

"Healthcare commissioners in England face a challenge in balancing competing needs. This paper contributes to our understanding of this by quantifying the bi-lateral associations, or trade-offs, with CYP MH spend when allocating budgets. Our main findings are that per capita spend on CYP MH was positively associated with need, the level of children in need stands out as significant among the variables selected; positively associated with spend on adult mental health; but negatively associated with spend on physical health and the proportion of the population which is from black and minority ethnic backgrounds."

Comment: My final comments on the modelling part is about the "other variables". The authors successfully included variables like unemployment rate to control for the variance of social

demographics between populations in different CCGs. However, some other environmental variables might be considered too, like price difference between CCGs (Martin et al., 2008).

CLARIFICATION: Thank you for the suggestion and paper cited. Indeed, the market forces factor (MFF) is already part of the allocation formula used by NHS England for setting CCG budgets. We have included additional variables in the list of confounders, such as the suggested MFF. The additional confounders are now listed in the methods section:

“...We also included the Market Forces Factor, an index that adjusts for variation in costs, as well as the percent of the population aged 65 years or more and 85 years or more. Finally, we included the rates of overcrowding of households and hospital stays for alcohol related harm.”

Comment: The modelling work is based on very small sample (under 200 data points) for one year only (FY 2016/17). Therefore, the results derived from this study cannot reflect accurate causality relationship but association relationship only. The authors might consider of making this point explicit to avoid confusions by readers with no strong statistical background.

DONE: We have now added an explicit disclaimer to that effect in the Strengths and Limitations section that now reads:

“Limited to a single year of spending, this paper can provide evidence only of association and not causality.”

VERSION 2 – REVIEW

REVIEWER	Ingrid Zechmeister-Koss Ludwig Boltzmann Institute for Health Technology Assessment, Austria
REVIEW RETURNED	12-Aug-2019

GENERAL COMMENTS	dear authors, I have read the revisions you made and my concerns and suggestions for revision have all been addressed satisfactorily.
---